# PolyVivid: Vivid Multi-Subject Video Generation with Cross-Modal Interaction and Enhancement

Teng Hu[1]*   Zhentao Yu[2]*   Zhengguang Zhou[2]   Jiangning Zhang[3]
Yuan Zhou[2]   Qinglin Lu[2]   Ran Yi[1]†
[1]Shanghai Jiao Tong University   [2]Tencent Hunyuan   [3]Zhejiang University
Project page: https://sjtuplayer.github.io/projects/PolyVivid

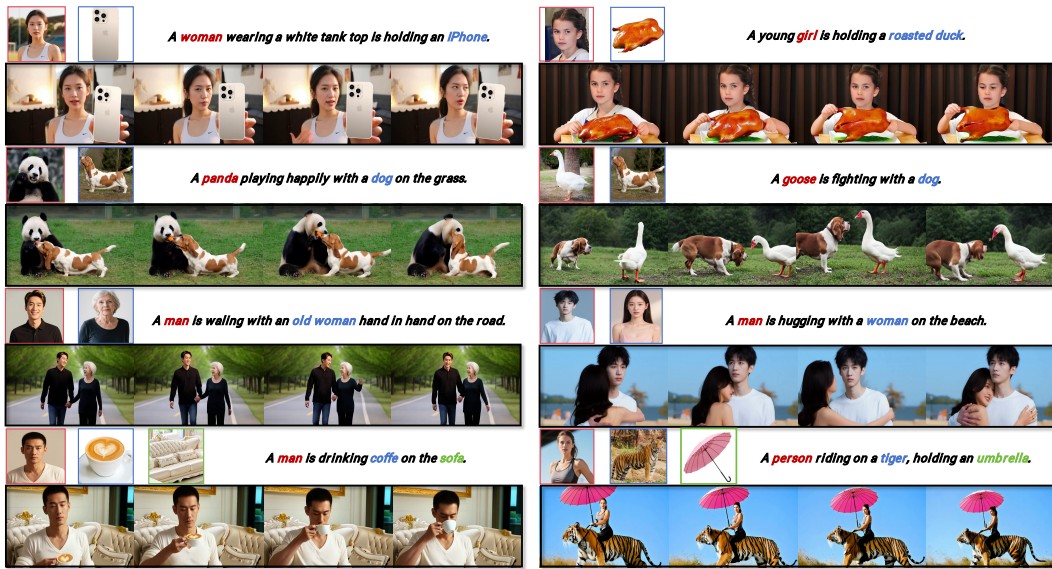

Figure 1: PolyVivid generates high-quality customized videos from multiple subject images and a text prompt, which ensures a high subject similarity and good subject interaction specified by text.

## Abstract

Despite recent advances in video generation, existing models still lack fine-grained controllability, especially for multi-subject customization with consistent identity and interaction. In this paper, we propose PolyVivid, a multi-subject video customization framework that enables flexible and identity-consistent generation. To establish accurate correspondences between subject images and textual entities, we design a VLLM-based text-image fusion module that embeds visual identities into the textual space for precise grounding. To further enhance identity preservation and subject interaction, we propose a 3D-RoPE-based enhancement module that enables structured bidirectional fusion between text and image embeddings. Moreover, we develop an attention-inherited identity injection module to effectively inject fused identity features into the video generation process, mitigating identity drift. Finally, we construct an MLLM-based data pipeline that combines MLLM-based grounding, segmentation, and a clique-based subject consolidation strategy to produce high-quality multi-subject data, effectively enhancing subject distinction and reducing ambiguity in downstream video generation. Extensive experiments demonstrate that PolyVivid achieves superior performance in identity

---

*Equal Contribution
†Corresponding author.

39th Conference on Neural Information Processing Systems (NeurIPS 2025).

fidelity, video realism, and subject alignment, outperforming existing open-source and commercial baselines.

# 1  Introduction

In recent years, the field of video generation has witnessed remarkable progress [46, 27, 18, 30, 17, 49, 8], with the emergence of numerous open-source and commercial video-generation models. These advancements have significant real-world implications, ranging from content creation in the entertainment industry to applications in artist design, education, advertising, etc. However, despite these achievements, current video-generation models suffer from a notable limitation – insufficient controllability. It remains challenging for these models to generate videos precisely tailored to users' specific requirements, which restricts their potential applications in various scenarios.

**Video customization** aims to generate videos featuring user-specified subjects. Existing approaches fall into two categories: (1) **Instance-specific methods** that fine-tune the model for each subject identity [50, 48, 23, 14], which are time- and resource-intensive; and (2) **End-to-end methods** that extract identity features from subject images and inject them into the generation process. While recent works like ConsisID [56] and MovieGen [38] show promise, they are limited to single-human scenarios and cannot handle arbitrary subject types or multiple identities.

To support **multi-subject video customization**, recent methods like ConceptMaster [21], Video Alchemist [7], Phantom [32], SkyReels-A2 [11], and VACE [24] extend single-subject frameworks by incorporating multiple image conditions. However, they still struggle with three key challenges: (1) learning precise correspondences between subject images and textual entities to model correct actions and interactions; (2) maintaining identity consistency across multiple subjects; and (3) resolving semantic ambiguity during data construction for accurate text-image alignment.

To address the above challenges, we propose **PolyVivid**, a novel multi-subject video customization framework that enables flexible, controllable, and identity-consistent video generation. To establish accurate correspondences between subject images and their textual descriptions, we first design a **VLLM-based text-image fusion module**. By leveraging the semantic understanding capabilities of Vision Large Language Models (VLLMs), this module encodes subject images into the textual embedding space, enabling the model to correctly ground each image to its corresponding entity in the prompt. To further enhance identity preservation and enable richer subject interactions, we propose a **3D-RoPE-based identity-interaction enhancement module**. This module enables structured bidirectional information flow between the text embeddings from VLLM and the subject image embeddings encoded by a pretrained VAE, enhancing identity information in text embeddings and injecting interaction semantics into the image embeddings. Finally, to efficiently inject both the identity-enhanced text embeddings and the interaction-enhanced image embeddings, we propose an **attention-inherited identity injection module**, which leverages the pretrained MM-Attentions multimodal processing capability to construct a new condition injection module. In this way, each frame can receive sufficient conditioning information, ensuring consistent subject appearance and preventing identity drift throughout the video. Additionally, to mitigate confusion between semantically similar entities, such as multiple humans, we develop a **MLLM-based data construction pipeline** that integrates MLLM-based grounding and segmentation with the proposed clique-based subject consolidation module. This pipeline leverages cross-modal cues to enhance subject discriminability and employs clique analysis to filter out inconsistent or transient subjects, further improving the accuracy and robustness of the generated results.

PolyVivid has been extensively experimented on multi-subject video customization. We compared it with existing open-source methods and closed-source commercial software, conducting comprehensive comparisons from multiple aspects such as ID consistency, authenticity of the generated videos, and video-text consistency. Extensive experiments show that PolyVivid outperforms all existing methods in multi-subject video customization. The contributions can be summarized in four-fold:

- We propose **PolyVivid**, a novel **multi-subject video customization model** that leverages VLLM models to establish the correspondences between the text prompt and subject images, enabling high subject-consistency video generation and complex subject interactions.
- We propose a new **3D RoPE-based identity-interaction enhancement module** to enable structured bidirectional information flow between textual and visual modalities. This mech-

anism enhances identity information in the text embedding and enriches the interaction information in the image embeddings, enabling more effective cross-modal interaction.

- We design an **Attention-inherited identity injection module** that utilizes the multimodal processing capabilities of the pretrained MM-Attention to develop a novel condition injection module. This approach ensures that each frame receives sufficient conditioning information, maintaining consistent subject appearance and preventing identity drift throughout the video.

- We propose a **MLLM-based data construction pipeline** that combines MLLM-based grounding, segmentation, and clique-based subject consolidation to improve multi-subject discriminability and reduce subject ambiguity in multi-subject data construction.

## 2 Related Work

### 2.1 Video Generation Model

The development of diffusion models [41, 16] has significantly advanced vision generation [52, 19, 55]. Early models [2, 12] extended pre-trained text-to-image models for continuous video generation by adding temporal modeling. Recently, works [34, 46, 27, 57, 53] have employed advanced Diffusion Transformers [36, 10], trained on large-scale text-video data, to produce longer and higher-quality videos. However, while many models focus on text-to-video and image-to-video generation, there is still potential for improving fine-grained controllability in video generation.

### 2.2 Video Customization

**Instance-specific video customization** methods use multiple images of the same subject to fine-tune pre-trained video generation model, training each subject separately. Still-Moving [4] fine-tunes video models with PEFT methods [15, 20] to create static frame videos, then repeats the image as a static video and uses DreamBooth [42] to learn the identity. CustomCrafter [50] repeats the image across frames, embeds it into text space, and fine-tunes the model for better identity learning. CustomVideo [48] and DisenStudio [5] extend customization to multiple subjects by segmenting and combining images, aligning subject identity with text through cross-attention maps. These methods rely on instance-specific optimization, posing challenges for real-time or large-scale video customization.

**End-to-end video customization** methods integrate identity information from target images through additional conditioning networks, which allows for the generalization to various identity inputs. Earlier works focus on preserving facial identity. For example, ID-Animator [14] uses a face adapter and facial identity loss to ensure facial ID consistency. ConsisID [56] captures comprehensive ID information by extracting low- and high-frequency details from facial images. MovieGen [38] embeds facial ID information into the text space and uses facial images from different videos to guide generation, reducing facial copying issues. To customize arbitrary objects, VideoBooth [23] incorporates identity information using coarse CLIP features and fine-grained image features. Recent works like ConceptMaster [21], Video Alchemist [7], Phantom [32], SkyReels-A2 [11], VACE [24], and HunyuanCustom [18] extend customization to multiple subjects by linking text prompts to subject images, enabling multi-subject video generation. However, challenges remain in maintaining and interacting with multiple subject IDs due to the complexity of interactions and mutual influence among them.

## 3 MLLM-based Data Construction

In this section, we outline the creation of our multi-subject customization dataset, emphasizing the grounding and segmentation process. Details on data filtering and captioning are in the appendix.

**MLLM-based Subject Segmentation.** For subject extraction in videos, previous methods either use GroundingDINO+SAM [33, 40] to detect object boxes based on text and then segment the boxes with SAM, or employ MLLM-based segmentors like LISA [28] to directly segment the subject parts corresponding to the captions. However, these methods have limitations: (1) The GroundingDINO+SAM approach struggles with fine-grained semantic distinctions, such as differ-

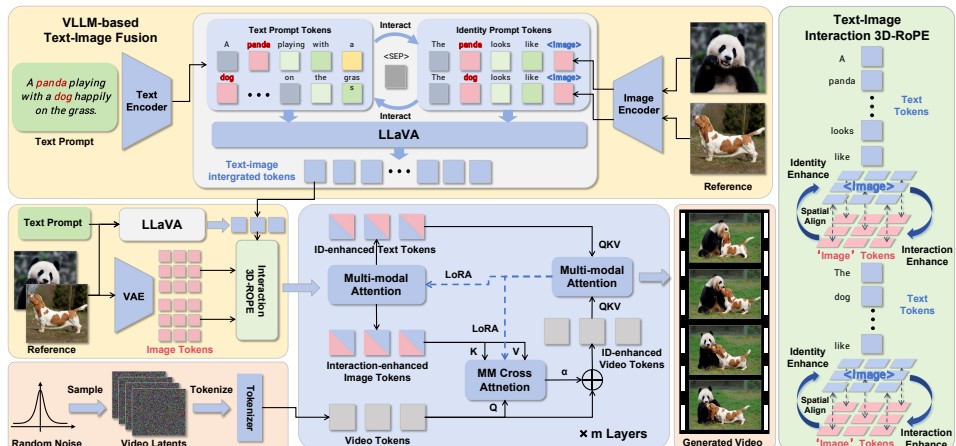

Figure 2: Framework of our PolyVivid: the text prompt and reference image are fused by the VLLM-based text-image fusion module. Then, a 3D RoPE-based identity-interaction enhancement module is employed to enhance the text-image interaction. The enhanced image tokens are injected by an MM cross-attention module, which helps preserve the identities while ensuring good subject interaction.

entiating between two people in a video, leading to inaccurate caption-to-semantic correspondence. (2) MLLM-based segmentation methods can better distinguish fine-grained semantics but suffer from fragmented and low-quality masks due to the scarcity of multimodal segmentation data. To address these issues, we adopt a MLLM-based detection+SAM segmentation paradigm for accurate and efficient multi-subject segmentation. Specifically, we use Florence2 [51], a MLLM-based detection model, to detect subject location boxes in a caption given an image and caption. We then use SAM2 to segment within these boxes, selecting the largest object as the subject. The segmentation is considered valid if the CLIP score between the segmented subject and the corresponding text exceeds a certain threshold.

**Clique-based Subject Consolidation.** Finally, to further enhance the accuracy of multi-subject detection and segmentation and prevent any subject from fleetingly appearing in the video, we extract CLIP features for each subject image and construct a graph $G = (V, E)$, where each node represents a subject image. We calculate the feature distance between each pair of subject images, connecting them with an edge if the distance is below a certain threshold. We then iteratively detect the maximum clique (a subgraph where every pair of nodes is connected) in the graph. If the number of nodes in the maximum clique exceeds one-third of the total detected frames, we extract it from the graph and continue searching for the next maximum clique, until the maximum clique has fewer nodes than one-third of the total detected frames. Ultimately, we obtain multiple cliques, each representing a reference image of a subject in the video. This algorithm effectively avoids errors in detection and segmentation in certain frames and removes subjects that appear in only a few frames, thereby improving the quality and robustness of the multi-subject generation model.

## 4 Method

PolyVivid is proposed for multi-subject video customization. It generates videos from multiple subject images $\{I_1, I_2, \cdots, I_n\}$ and a text prompt describing scenes, actions, and interactions, ensuring identity preservation and accurate text-specified interactions. The framework is shown in Fig. 2. First, a **VLLM-based text-image fusion module** encodes both text and subject images into the text space as $\{z_{T,T}, z_{T,I}\}$, capturing high-level semantic associations. To address LLaVA's limitation in fine-grained identity details, we use a pretrained VAE to extract detailed visual identity features from the subject images, obtaining image embeddings $z_I$. Next, an **identity-interaction enhancement module** with **text-image interaction 3D RoPE** facilitates mutual information flow between modalities. Image tokens inject identity information into text tokens, while text tokens provide interaction cues to image tokens, resulting in identity-enhanced text tokens and interaction-aware image tokens. Finally, an **Attention-Inherited Identity Injection Module** injects interaction-enhanced image tokens into the video latent space via an MM cross-attention mechanism to ensure consistent identity preservation, and the identity-enhanced text tokens guide subject-specific interactions during video

generation using the pretrained MM-attention. By integrating these components, PolyVivid achieves high-fidelity video generation with identity-consistent and text-aligned multi-subject interactions.

## 4.1 VLLM-based Text-image Fusion

In video customization tasks, the model first needs to learn the relationship between the input text and images to identify which subject image corresponds to which entity in the text, enabling the generation of corresponding subject actions as described in the prompt. However, how to effectively integrate image-text information has been a key challenge for previous customization methods. These methods usually take subject image features and text embeddings as two separate conditions, and either lack a design for interactive understanding between them (lacks the learning of correspondence between text entities and subject images), or rely on additional newly trained network branches to achieve this interaction, causing the model prone to confusing different subjects within the generated video. To facilitate efficient image-text interaction understanding, PolyVivid leverages the text comprehension capabilities trained in the LLaVA [31] text space by HunyuanVideo [27] and utilizes LLaVA's multimodal interaction understanding to build the connection between the text and images [43, 29].

Given a text input $T$ and several subject images $\{I_1, \cdots, I_n\}$, each with a corresponding description word $\{T_{I,1}, \cdots, T_{I,n}\}$ in the text, we design a **structured template** to explicitly link each image with its textual entity. This template is processed by LLaVA to learn multi-modal correlations between text and image identities. LLaVA is a multi-modal model trained for visual question answering and dialogue tasks. It uses a conversation-style prompt where images and texts are interleaved, employing a pretrained vision encoder (i.e., CLIP-ViT) to extract image features, which are integrated with text tokens through causal language modeling. Our structured template appends identity prompts after the input text prompt, associating each subject word with its corresponding subject image, formatted as:

$$\underbrace{T}_{\text{Text prompt}} \texttt{<SEP>} \underbrace{\text{The } T_{I,1} \text{ looks like } \texttt{<image 1>}.}_{\text{Identity prompt 1}} \underbrace{\text{The } T_{I,2} \text{ looks like } \texttt{<image 2>}.}_{\text{Identity prompt 2}} \cdots \quad (1)$$

where `<SEP>` is the special token marking the end of a dialogue round; `<image i>` is the token for subject image $i$. For example, with the text prompt "A man is playing guitar", the resulting template is "A man is playing guitar. `<SEP>` The man looks like `<image 1>`. The guitar looks like `<image 2>`".

When the template is input to LLaVA, each `<image i>` token is replaced with visual tokens extracted from the corresponding image using a CLIP image encoder, typically resulting in $24 \times 24$ tokens per image. This structured template is then processed by LLaVAs autoregressive language model to produce joint multi-modal embeddings. Due to the much longer sequence of image tokens compared to text tokens, directly appending them can cause the model to be overly influenced by visual content, disrupting textual understanding. To address this, the insertion of `<SEP>` acts as a soft delimiter, reducing interference between text and image parts while allowing LLaVA to correctly associate subjects with their visual references. After LLaVA processes the multimodal image-text interaction, the output is fused text embeddings $z_T$, which can serve as text inputs for the video generation model to synthesize customized videos that reflect the specified subjects and contextual information.

## 4.2 Identity-Interaction Enhancement by Text-image Interaction

The LLaVA model used in our text-image fusion (Sec. 4.1, as a multimodal understanding framework, is primarily designed to capture high-level semantic correlations between text and imagessuch as category, color, and shapewhile often overlooking finer-grained details like textural cues and detailed visual attributes. However, in the context of video customization, these fine details are crucial for accurate identity preservation, making the LLaVA branch alone insufficient. To complement this, we leverage the pretrained VAE from the base video generation model (HunyuanVideo) to encode subject images into image embeddings $z_I$, which effectively retain identity-specific information. So far, we have obtained a text embedding $z_T$ from LLaVA (Sec. 4.1) that captures interaction-related semantics but lacks detailed identity information, and an image embedding $z_I$ that captures identity features but lacks subject interaction context. To compensate for these missing information, we pro-

pose a text-image interaction module based on 3D-interacted RoPE, designed to enhance identity features in $z_T$ and enhance interaction semantics in $z_I$.

**Text-image Interaction Module.** Our base video generation model (HunyuanVideo) is built upon a Multimodal Attention Framework, where each model block contains a MM-attention module that integrates text and video embeddings to establish cross-modal interactions and achieve video-text alignment. Since a subject image can be viewed as a single-frame video, we repurpose the MM-attention mechanism to facilitate interaction between text and image embeddings. This enables the injection of identity information from image embedding $z_I$ into text embedding $z_T$ and, conversely, the infusion of semantic interaction cues from text embedding $z_T$ into image embedding $z_I$. Specifically, the MM-attention module in the base model contains the *Query*, *Key*, and *Value* matrices for both video embedding $V$ and text embedding $T$, *i.e.*, $W_{\{q,k,v\}}^V$ for video embedding, and $W_{\{q,k,v\}}^T$ for text embedding. To effectively process the image and text embeddings, we adopt a LoRA [15]-based approach to fine-tune the *Query*, *Key*, *Value* matrices, and feed-forward network (FFN) weights, enabling efficient and effective adaptation of the MM-attention layers for text-image fusion:

$$z_I', z_T' = MM\text{-}Attn(\{\hat{W}_q^V(z_I), \hat{W}_q^T z_T\}, \{\hat{W}_k^V(z_I), \hat{W}_k^T z_T\}, \{\hat{W}_v^V(z_I), \hat{W}_v^T z_T\}),$$
$$\hat{z}_I, \hat{z}_T = F\hat{F}N(z_I'), F\hat{F}N(z_T'), \tag{2}$$

where $\{\hat{W}_q^{\{V,T\}}, \hat{W}_k^{\{V,T\}}, \hat{W}_v^{\{V,T\}}, F\hat{F}N\}$ are the weights combined by the pretrained MM-Attention and a LoRA module.

**Text-image Interaction 3D-RoPE.** Our text-image interaction module takes both text embedding and image embedding as inputs, aiming to enhance identity information in the text embedding and enrich interaction semantics in the image embedding. However, directly integrating these modalities through a standard attention mechanism often fails to capture their mutual correspondence effectively, as text tokens are organized as one-dimensional sequences, while image tokens exhibit a two-dimensional spatial structure. This mismatch hinders the alignment and fusion of information between modalities, leading to ineffective interaction modeling. To overcome this challenge, we propose a **text-image interaction 3D-RoPE**, which facilitates structured and fine-grained positional encoding for both modalities, in order to bind text tokens and image tokens of the same subject. This design enables more effective cross-modal interaction while preserving the intrinsic semantics of each modality.

Specifically, the text embedding $z_T$ consist of two types of tokens: $z_{T,T}$ and $z_{T,I}$, where $z_{T,T}$ is derived from the input text prompt and $z_{T,I}$ is derived from the input subject images. The text tokens are represented as:

$$z_T = \{\underbrace{z_{T,T}^1, \cdots, z_{T,T}^{m_1}}_{\text{text}}, \underbrace{z_{T,I}^{m_1+1}, \cdots, z_{T,I}^{m_2}}_{\text{<image 1>}}, \underbrace{z_{T,T}^{m_2+1}, \cdots, z_{T,T}^{m_3}}_{\text{text}}, \underbrace{z_{T,I}^{m_3+1}, \cdots, z_{T,I}^{m_4}}_{\text{<image 2>}}\}. \tag{3}$$

We denote the text tokens $z_{T,T}$ encoded from the text prompt as <text> tokens, and the text tokens $z_{T,I}$ encoded from the subject image as <image> tokens. In addition, the 'image' tokens $z_I$ refer to the tokens extracted from the subject image by the VAE encoder. The differences between these two kinds of image tokens are that: The <image> tokens $z_{T,I}$ (obtained from LLaVA) reside in the text space and capture rich interaction information derived from the text, while the 'image' tokens $z_I$ (obtained from VAE encoder) retain detailed identity features extracted from the subject image.

Our text-image interaction 3D-RoPE is shown in Fig. 2. To preserve the source text information, it is crucial to maintain the sequential relationship in $z_{T,T}$. Therefore, we assign the <text> tokens in $z_{T,T}$ with position encodings of 3D RoPE, setting the spatial dimension to $(0, 0)$, and assign them sequentially along the temporal dimension. Take the first subject as an example, the RoPE position indices for its <text> tokens are set as:

$$Idx_{RoPE}(z_{T,T}^i) = (i, 0, 0), \quad i = 1, \cdots, m_1. \tag{4}$$

This method preserves the original sequential relationship of $z_{T,T}$ along the temporal axis. To model the interaction between <text> tokens and <image> tokens, we assign the <image> tokens with 3D RoPE of temporal index $m_1 + 1$ and expanded along the spatial axis, centering them at $(0, 0)$ in the spatial dimensions. The RoPE position indices for the first subject's <image> tokens are set as:

$$Idx_{RoPE}(z_{T,I}^{m_1+i}) = (m_1 + 1, \lfloor \frac{i}{h} \rfloor - \lfloor \frac{w}{2} \rfloor, i \bmod h - \lfloor \frac{h}{2} \rfloor), \quad i = 1, \cdots, wh. \tag{5}$$

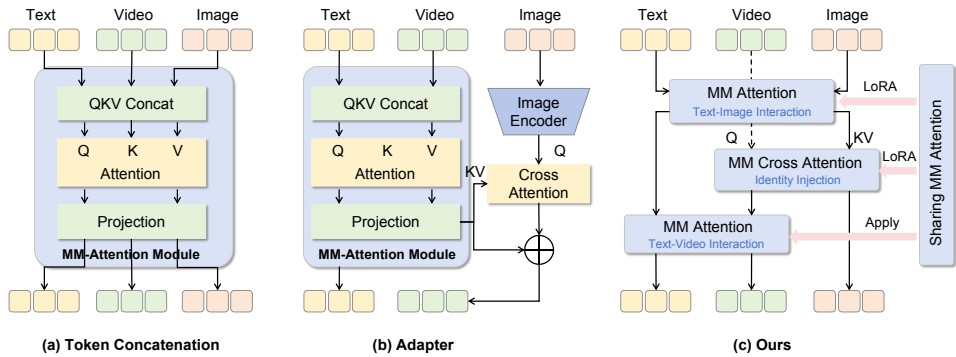

**Figure 3:** Comparison of the condition injection strategies for MM-DiT.

where $w$ and $h$ are the width and height of the encoded image, respectively. Since the same subject's <text> tokens' and <image> tokens' RoPE positions are close in temporal dimension, their position encodings are close, thereby stronger correlations are more likely to be obtained in MM-Attention.

For the 'image' tokens $z_I$, we assign them with 3D RoPE of temporal index $m_1 + 2$ and spatial indices aligned with those of the <image> tokens. This alignment facilitates efficient pixel-by-pixel interaction between the two sets of image tokens. The RoPE position indices for the first subject's 'image' tokens $z_I$ are set as:

$$Idx_{RoPE}(z_I^i) = (m_1 + 2, \lfloor \frac{i}{h} \rfloor - \lfloor \frac{w}{2} \rfloor, i \texttt{ mod } h - \lfloor \frac{h}{2} \rfloor), \quad i = 1, \cdots, wh, \tag{6}$$

For multi-subject inputs, we expand them iteratively along the time axis, with the second subject's <text> tokens starting from $(m_1 + 3, 0, 0)$. The relationships among <text>, <image>, and 'image' tokens remain consistent as described in Eq. (4) to Eq. (6).

This text-image 3D RoPE effectively preserves the sequential relationship between <text> and <image> tokens. Additionally, it enables efficient pixel-by-pixel interaction between <image> tokens and 'image' tokens, enhancing the identity of the <image> tokens. Furthermore, since 'image' tokens are embedded between text tokens through this 3D RoPE, they can efficiently interact with text tokens, thereby enriching the interaction information in 'image' tokens. With this text-image interaction 3D-RoPE, it enables structured bidirectional information flow between text and image tokens via Eq. (2), resulting in identity-enhanced text tokens $\hat{z}_T$ and interaction-enhanced image tokens $\hat{z}_I$.

### 4.3 Attention-inherited Identity Injection

With the identity-enhanced text tokens $\hat{z}_T$ and interaction-enhanced image tokens $\hat{z}_I$ obtained in Sec. 4.2, the remaining challenge is how to effectively inject both types of tokens into the video generation process to produce subject-consistent videos aligned with the text-described context.

**Problems in Current Controllable Generation Methods.** Our model is built upon HunyuanVideo, which is based on MM-DiT, and we consider two common condition injection strategies used in MM-DiT: **1) Token concatenation** (Fig. 3 (a)), as used in OmniControl [44], involves concatenating the condition tokens with the latent video tokens and using MM-Attention to learn their interaction. **2) Adapter-based methods** (Fig. 3 (b)), such as IP-Adapter [54], extract condition features via an external image encoder and inject them into the pretrained model through cross-attention layers. Both approaches are originally designed for image generation, and applying them directly to video generation presents challenges. In token concatenation, appending image tokens before or after video tokens can lead to temporal imbalance, *i.e.,* frames distant from the condition image receive weaker guidance, resulting in identity degradation. Adapter-based methods struggle to inject information effectively into MM-DiT due to the large dimensionality of its feature space (considering the long token sequence of video); the adapter's features often misalign with the pretrained model's internal representations, hindering effective conditioning.

**Attention-inherited Identity Injection.** To overcome the limitations of existing controllable generation methods, we propose a novel *Attention-Inherited Identity Injection Module*. The "Attention-Inherited" principle refers to a parameter-sharing mechanism designed for efficiency and performance. At its core, we utilize a single set of MM-Attention parameters inherited from the pretrained base model. This foundational weight set is shared across three key components: the text-image

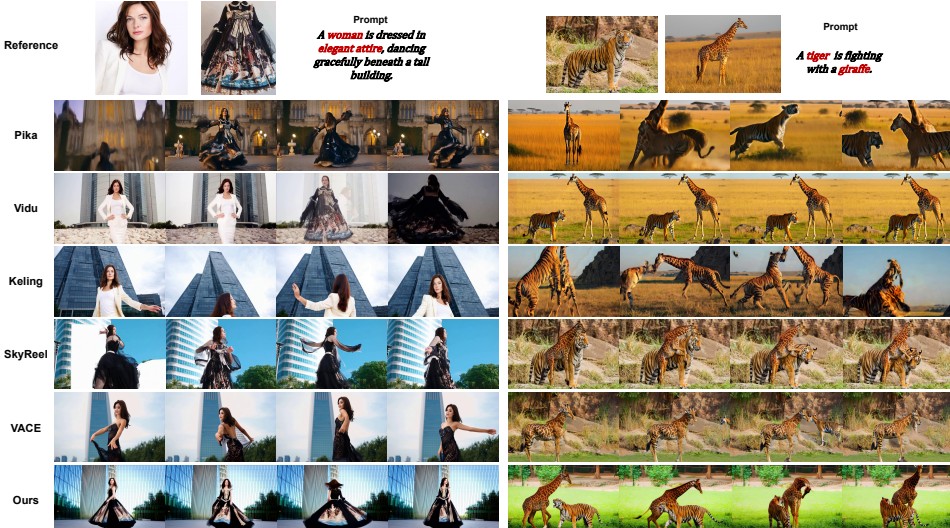

Figure 4: Comparison on multi-subject video customization.

attention, the text-video attention, and our identity injection cross-attention module (see Fig. 3 (c)). We reparameterize the *Key* and *Value* matrices of the video tokens using a LoRA module to incorporate image token information, and reparameterize the *Query* matrices using another LoRA module to maintain video-specific representations. This setup constructs a **multi-modal cross-attention mechanism** where image tokens $\hat{z}_I$ are injected into the video token stream $z$, ensuring effective identity injection. This design treats all video frames equally, eliminating disparities between earlier and later frames of the token concatenation strategy, thus mitigating identity degradation over time. To stabilize early-stage training, we use a zero-initialized fully-connected layer to project the cross-attention output, reducing the impact of randomly initialized attention weights. The injection process is formulated as:

$$\hat{z}_I, \hat{z}_T = F\hat{F}N(MM\text{-}Attn_{T,I}(z_I, z_T)) \tag{7}$$

$$z' = Cross\text{-}Attn(\hat{W}_q^V(z), \hat{W}_k^V(\hat{z}_I), \hat{W}_v^V(\hat{z}_I)), \tag{8}$$

$$\hat{z} = z + FC(F\hat{F}N(z')), \tag{9}$$

$$\hat{z}_{out}, \hat{z}_{T,out} = MM\text{-}Attn_{T,V}(\hat{z}, \hat{z}_T) \tag{10}$$

where $MM\text{-}Attn_{T,I}$ and $MM\text{-}Attn_{T,V}$ are the Multi-Modal Attention modules for text-image and text-video interaction, respectively. The parameters $\{\hat{W}_q^V, \hat{W}_k^V, \hat{W}_v^V, F\hat{F}N\}$ are composed of weights from the pretrained MM-Attention and a LoRA module. $FC(\cdot)$ is a zero-initialized fully connected layer, and $\hat{z}$ denotes the identity-enhanced video tokens.

After the identity injection, we utilize the original MM-Attention module from the pretrained model to establish the connection between the identity-enhanced text tokens $\hat{z}_T$ and the identity-enhanced video tokens $\hat{z}$. This allows the text information to be effectively integrated into the video tokens without compromising identity integrity, thereby supporting both strong identity preservation and vivid subject interaction generation.

## 5 Experiments

### 5.1 Implementation Details

**Baselines.** We compare PolyVivid with the state-of-the arts video customization methods, including commercial products (Vidu-2.0 [45], Kling-1.6 [25], Pika [37], and Hailuo [13]) and open-sourced methods (Skyreels-A2 [11] and VACE-1.3B [24]). For each model, we generate 100 videos, which are employed to compute the quantitative metrics. More implementation details are presented in the appendix.

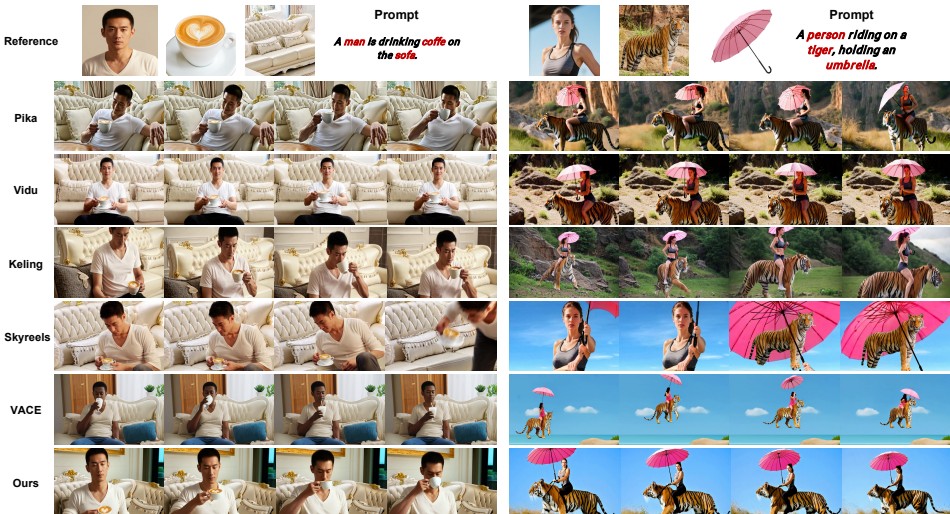

Figure 5: Comparison of three-subject customization.

## 5.2 Comparison Results on Multi-subject Customization

**Qualitative two-subject customization comparisons.** We conduct experiments on four types of multi-subject customized video generation to evaluate the effectiveness of our model: **(1)** Rigid human-object interaction (e.g., a person holding an object); **(2)** Non-rigid human-object interaction (e.g., a person wearing flexible clothing); **(3)** Human-human interaction; and **(4)** Object-object interaction. In Fig. 4 (with Rigid human-object interaction and Human-human interaction included in the appendix), we compare our model with the state-of-the-art methods. It can be seen that Pika often produces blurry outputs when handling complex interactions. Vidu and Kling struggle to dress the human subjects in the specified clothing. Vidu and VACE confuse animal features, generating a tiger with the shape of a giraffe or a giraffe with the shape of a tiger. SkyReels-A2 introduces noticeable artifacts and inconsistent transitions between frames, and struggles to accurately capture the relative size of different animals (the generated tiger is bigger than the giraffe). Furthermore, all baseline methods except Kling suffer from significant identity degradation, especially in humans. In contrast, our PolyVivid generates videos with high identity consistency while effectively modeling complex interactions between multiple subjects as specified by the text prompts.

**Qualitative three-subject customization comparisons.** Our model is not limited to two-subject customization. We present additional comparison results for three-subject video customization in Fig. 5. As shown, Pika, Vidu, Skyreels A2, and VACE all exhibit significant identity loss. While Pika and Vidu are able to generate correct interactions that adhere to physical rules, Skyreels A2 and VACE produce unrealistic frames in which the person and tiger appear pasted into the sky, violating physical plausibility. Kling maintains a relatively good identity preservation, but there is still room for improvement. In contrast, our model achieves the best identity preservation and is able to generate realistic interactions among multiple subjects while following physical rules, demonstrating our superior capability in multi-subject customization.

**Quantitative Comparisons.** To further demonstrate the effectiveness of our approach, we construct a benchmark test set containing 100 object pairs, each associated with a corresponding interaction text prompt. We apply each baseline method to generate 100 videos and evaluate the results using a comprehensive set of quantitative metrics, including face and object similarity, text-video alignment, and overall video quality. The comparative results are summarized in Tab. 1. As shown in the table, PolyVivid achieves the highest similarity scores for both face and object identity (Face-sim and DINO-sim), highlighting its strong capability in preserving key appearance features across video frames. In terms of text-video alignment, our method obtains the best and second-best CLIP scores among all competitors, suggesting that the generated content accurately reflects the intended semantics of the text prompts. Furthermore, PolyVivid yields the lowest FVD score, indicating superior video quality and diversity. It is worth noting that the FVD is computed against a reference set of 1,500 high-quality 4K videos, further underscoring the realism of our results. While our method ranks third in temporal consistency, we observe that certain videos generated by Vidu and VACE remain mostly static, which slightly inflates their temporal consistency scores. Despite this, our

Table 1: We compare PolyVivid with state-of-the-art video customization methods across multiple metrics. **Bold** and underline represent the best and second best results, respectively.

| Metric | Face-sim ↑ | DINO-sim ↑ | CLIP-B ↑ | CLIP-L ↑ | FVD ↓ | Temporal ↑ |
|---|---|---|---|---|---|---|
| VACE-1.3B [24] | 0.433 | 0.598 | 0.335 | 0.280 | 1171.42 | 0.966 |
| SkyReels-A2 [11] | 0.554 | 0.619 | 0.332 | 0.276 | 1379.65 | 0.943 |
| Kling-1.6 [25] | 0.534 | 0.554 | 0.330 | 0.280 | 1049.70 | 0.934 |
| Vidu-2.0 [45] | 0.532 | 0.588 | **0.336** | **0.282** | 1083.35 | **0.970** |
| Pika [37] | 0.546 | 0.548 | 0.310 | 0.263 | 980.49 | 0.942 |
| **PolyVivid (Ours)** | **0.642** | **0.623** | **0.336** | 0.281 | **959.74** | 0.964 |

Table 2: Quantitative Ablation Study.

| LLaVA | Text-Img Interaction | Text-Img 3D-RoPE | ID-injection | Face-sim↑ | DINO-sim ↑ | CLIP-B ↑ | CLIP-L ↑ | FVD ↓ | Temporal ↑ |
|---|---|---|---|---|---|---|---|---|---|
| ✓ | | | | 0.381 | 0.521 | **0.345** | **0.291** | 1150.48 | 0.950 |
| | ✓ | | | 0.345 | 0.496 | 0.334 | 0.280 | 1052.34 | 0.961 |
| ✓ | ✓ | | | 0.584 | 0.581 | 0.330 | 0.279 | 1052.14 | **0.965** |
| ✓ | ✓ | ✓ | | 0.601 | 0.605 | 0.340 | 0.286 | 965.62 | 0.963 |
| ✓ | ✓ | ✓ | ✓ | **0.642** | **0.623** | 0.336 | 0.281 | **959.74** | 0.964 |
| LLaVA + Adapter | | | | 0.401 | 0.532 | 0.338 | 0.285 | 1020.35 | 0.952 |
| LLaVA + Token-Concatenation | | | | 0.628 | 0.615 | 0.328 | 0.271 | 980.56 | 0.960 |
| Text-Only LLaVA + Token-Concatenation | | | | 0.543 | 0.532 | 0.330 | 0.267 | 1132.12 | 0.959 |

method still demonstrates strong temporal coherence. In summary, PolyVivid not only delivers the best performance in keeping identities across humans and objects, but also achieves strong semantic alignment and generation quality, validating its effectiveness for multi-subject video customization.

## 5.3 Ablation Study

In this section, we first compare our model with two condition injection strategies: (1) adapter-based injection and (2) token concatenation-based injection. For token concatenation, we additionally compare with an ablated model with text-only LLaVA (i.e., without text-image fusion). We then conduct an ablation study on four key components of our framework: (1) the text-image fusion module based on LLaVA, (2) the text-image interaction module, (3) the text-image interaction 3D-RoPE, and (4) the identity injection module. Quantitative results are shown in Tab. 2. The adapter-based model struggles to capture identity information, yielding only marginal improvements over the baseline with LLaVA alone. The token concatenation approach maintains identity better, but at the cost of reduced text-image alignment, as reflected by a lower CLIP score. Moreover, the model with text-only LLaVA results in poor identity consistency due to the limitation in distinguishing different identities. Individually, the LLaVA fusion and text-image interaction module do not provide strong identity preservation, but their combination leads to a notable improvement. Incorporating our proposed text-image interaction 3D-RoPE further enhances both identity consistency and text alignment, demonstrating its effectiveness for text-image interaction. Finally, the identity injection module significantly boosts identity preservation and achieves the best FVD score, confirming its effectiveness in subject-consistency video generation. By integrating all proposed modules, our method achieves strong identity preservation, accurate text-video alignment, and high-quality video generation.

## 6 Conclusion

In this paper, we present PolyVivid, a novel multi-subject video customization framework that addresses the key limitations of existing methods in controllability, identity consistency, and complex subject interaction. By incorporating a VLLM-based vision-language fusion module, a 3D-RoPE-based identity-interaction enhancement module, and an attention-inherited identity injection module, our model effectively bridges the gap between text and image modalities while preserving subject identities throughout video generation. Furthermore, our proposed multi-subject data construction pipeline enhances the ability to distinguish semantically similar entities, ensuring reliable multi-subject customization. Extensive experiments demonstrate that PolyVivid significantly outperforms prior state-of-the-art methods in identity consistency, text alignment, and video realism, achieving superior performance in multi-subject video customization, which opens up new possibilities for controllable and high-fidelity video generation in real-world applications.

## Acknowledgements

This work was supported by National Natural Science Foundation of China (No. 62302297), the Fundamental Research Funds for the Central Universities (YG2023QNB17, YG2024QNA44), National Key R&D Program of China (2024YFE0115500), National Natural Science Foundation of China (No, 72192821, 62272447, 62472282, 62472285), Young Elite Scientists Sponsorship Program by CAST (2022QNRC001), Beijing Natural Science Foundation (L222117).

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

# Appendix

## A Overview

In this appendix, we offer further details on implementation, present additional experimental results, and provide more comprehensive analyses, structured as follows:

- Implementation details (Sec. B);
- Multi-modal data curation (Sec. C);
- More quantitative evaluations (Sec. D);
- More multi-subject comparison results (Sec. E).
- More visualization results (Sec. F)
- Limitations and societal impacts (Sec. G)

## B Implementation details

**Progressive training process.** To enhance the efficiency of the training process, we divide it into two distinct stages. The first stage focuses on modeling the identity preservation capability, while the second stage targets the modeling of interaction generation. During the initial stage, the model is trained on single-subject data, concentrating solely on learning the target identity information without the complexity of interactions among multiple subjects. This stage involves 5,000 iterations. Once the model has effectively learned identity preservation, we proceed to the second stage, where the model is trained with multiple subjects as inputs. Here, the objective is to learn the interactions between the given subject images while maintaining their identities. This stage also comprises 5,000 iterations. Additionally, due to the extensive number of parameters in the pretrained HunyuanVideo model [27], each training iteration is time-consuming. To address this, in each stage, we initially train the model at reduced sizes for 1,000 iterations (included in the total 5,000 iterations), allowing the model to efficiently grasp the target objectives in less time. Subsequently, for the remaining iterations, we revert to the standard resolution to ensure the quality of the final output. All training processes are conducted on 256 GPUs, each with more than 80GB of memory, using a batch size of 256.

**Evaluation Metrics.** To comprehensively assess the performance of video customization, we adopt several metrics focusing on identity preservation, text-video alignment, and overall video quality:

- **Identity Similarity.** We utilize Arcface [9] to extract facial embeddings from both the reference image and each frame of the generated video, and then compute the average cosine similarity to evaluate how well the identity is preserved.

- **Subject Similarity.** Each frame is segmented using YOLOv11 [26] to isolate the subject, after which DINO-v2 [35] features are extracted. The similarity between these features and those from the reference is calculated to assess subject consistency.

- **Text-Video Alignment.** We employ CLIP-B and CLIP-L [39] to measure the correspondence between the provided text prompt and the generated video, evaluating how accurately the video reflects the textual description.

- **Fréchet Video Distance (FVD).** To assess the quality and diversity of the generated videos, we compute the FVD between generated and real videos. Video features are extracted using I3D [3], and the Fréchet Distance is then calculated.

- **Temporal Consistency.** Following the approach in VBench [22], we use the CLIP model to compute the similarity between each frame and its adjacent frames, as well as between each frame and the first frame, to evaluate the temporal coherence of the video.

**Test Dataset.** We manually collected 100 images of various objects, covering a wide range of categories such as man-made machines, food, goods, and buildings. In addition, we generated 100 human images using an image generation model. These images were then randomly paired to form 100 image pairs. For each pair, we utilized QWen2.5-VL [1] to generate corresponding interaction text prompts.

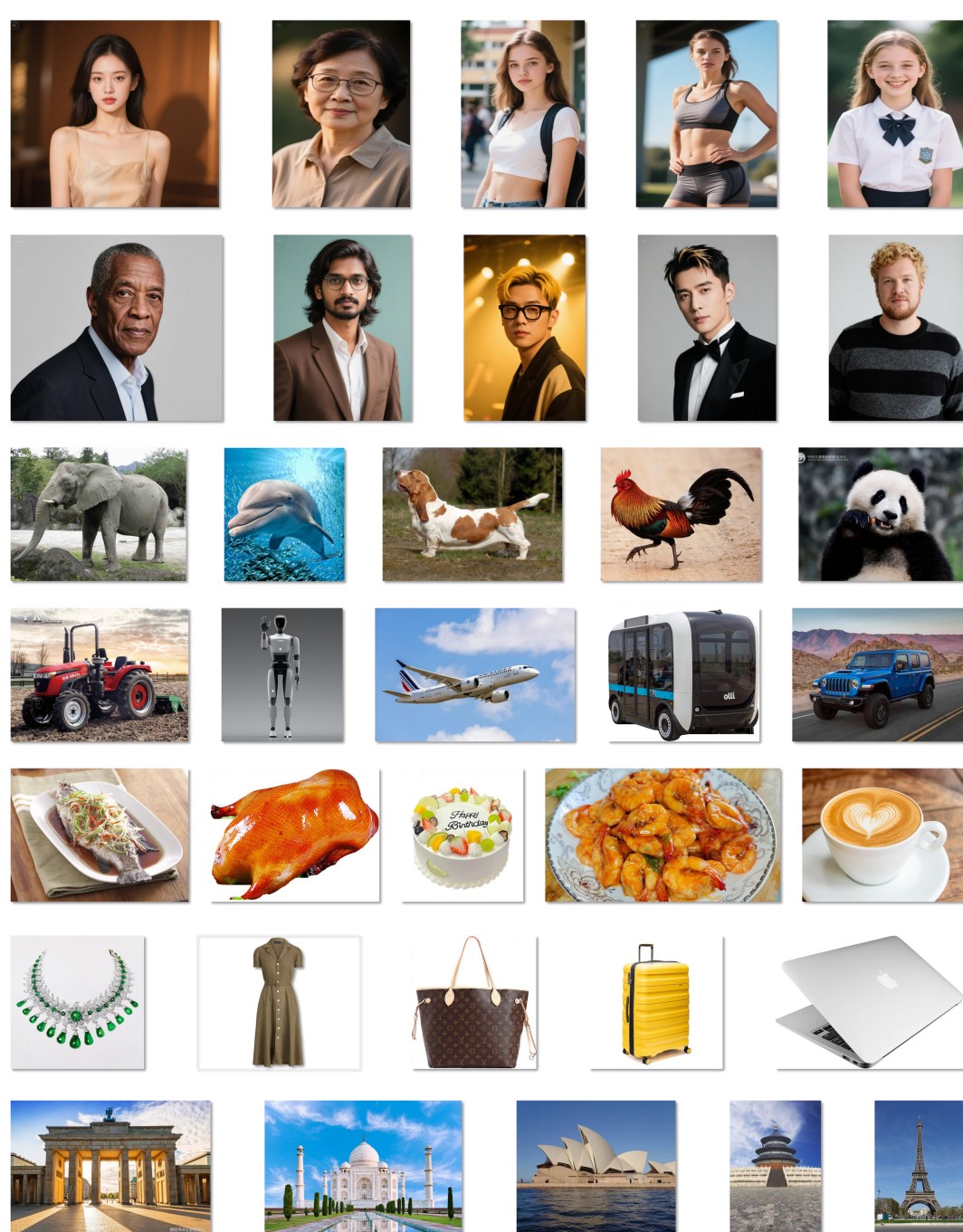

Figure A1: Examples of the test set, which contains images from diverse categories, such as human, animal, man-made machine, food, goods, and building.

## C   Multi-subject data curation

In the main paper, we have illustrated the MLLM-based Subject Segmentation stage and the Clique-based Subject Consolidation. In this section, we give more details for the preprocessing process, including the data source, data filtering and video captioning.

We curate a large set of high-quality data from open-source datasets, including Panda-70M [6] and Koala-36M [47], as well as our own collected data. Initially, we split the videos by dividing long

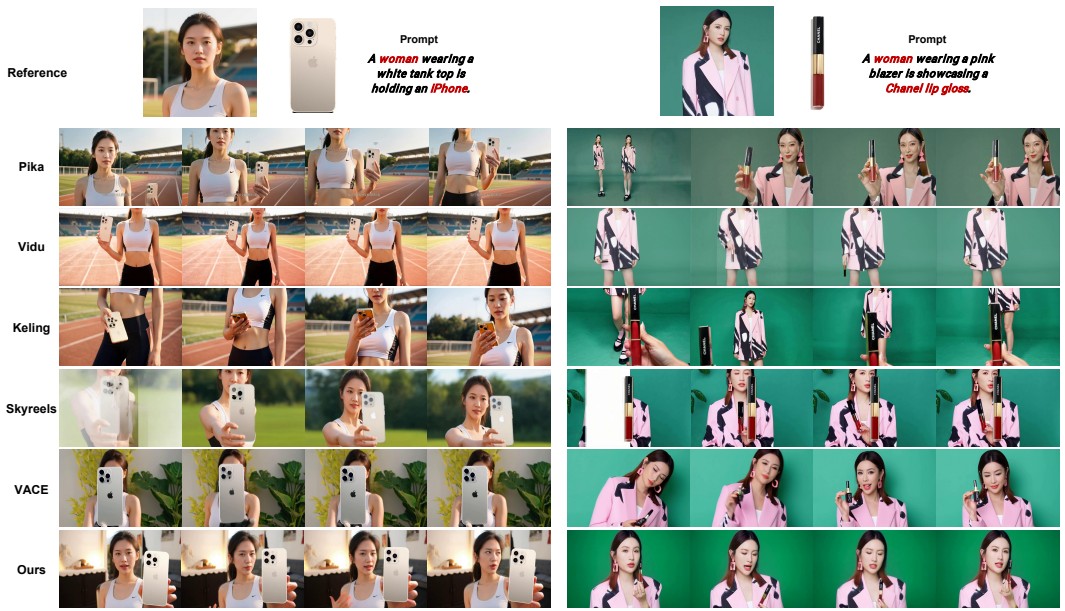

Figure A2: Comparison on human-object customization.

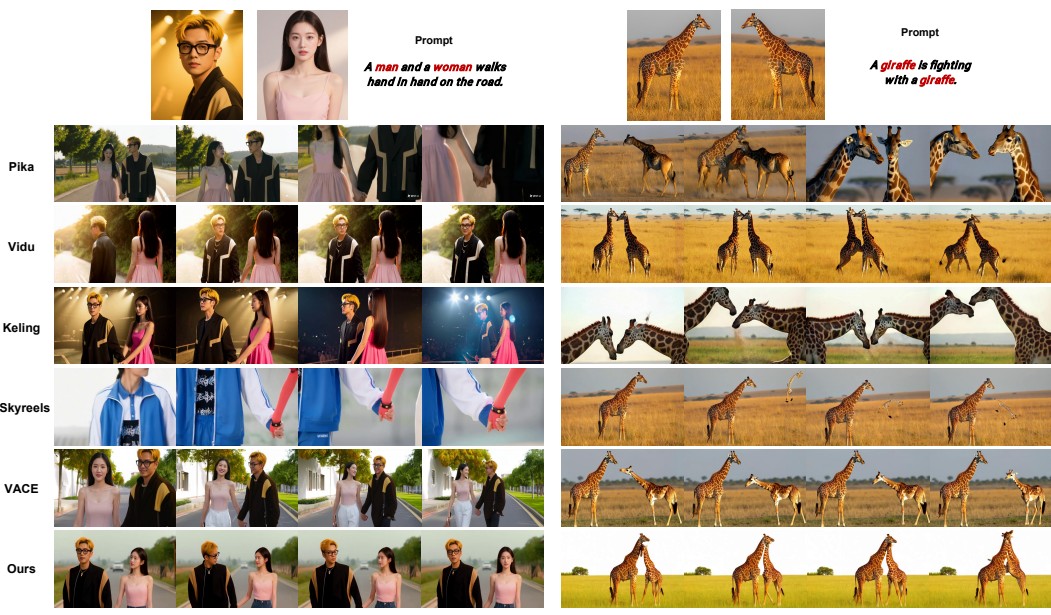

Figure A3: Comparison on human-human and animal-animal customizations.

videos into shorter clips. We then perform black border detection, subtitle detection, watermark detection, transition detection, and motion detection on these clips. Videos with black borders are cropped, and those with subtitles, watermarks, transitions, or low motion are removed. Further, we utilize Koala-36M [47] to filter out videos with scores below a certain threshold. We then perform structured video captioning on the remaining videos, generating long captions, short captions, and descriptions of background, style, and camera movement for each video. This structured combination is used during training to enhance caption diversity.

# D  More quantitative evaluations

For a more thorough and fine-grained evaluation of video generation quality, we conduct an expanded analysis using the VBench benchmark [22]. Our primary evaluation focused on Temporal Consistency, which encompasses subject and background consistency. To broaden this assessment, we now also include results for **Motion Consistency** and overall **Image Quality**, offering deeper insights into our model's capabilities.

Table A1 summarizes the performance of our method against several leading models. The results show that our model is highly competitive in maintaining temporal and motion consistency, achieving scores comparable to the best-performing methods. Notably, our model achieves a state-of-the-art score of **0.751** in Image Quality, demonstrating its superior ability to generate visually pleasing and high-fidelity video content. This highlights a key strength of our approach in balancing motion dynamics with perceptual quality.

Table A1: Comprehensive VBench evaluation. Our method demonstrates competitive consistency metrics and achieves the highest score in Image Quality, indicating superior visual fidelity. Best results are in **bold**, second best are underlined.

| Method | Temporal Consistency ↑ | Motion Consistency ↑ | Image Quality ↑ |
|---|---|---|---|
| Pika [37] | 0.942 | 0.995 | 0.729 |
| SkyReels-A2 [11] | 0.943 | 0.994 | 0.731 |
| VACE-1.3B [24] | 0.966 | 0.994 | 0.732 |
| Kling-1.6 [25] | 0.934 | 0.994 | 0.724 |
| Vidu-2.0 [45] | **0.970** | 0.997 | 0.695 |
| Ours | 0.964 | 0.995 | **0.751** |

# E  More multi-subject comparison results

**Human-object customization.** The ability to generate videos depicting human-object interactions is crucial, with broad applications in fields such as film production and advertising. We present qualitative results of human-object interaction in Fig. A2, where our method is compared against several state-of-the-art approaches, including Pika [37], Kling1.6 [25], Vidu2.0 [45], Skyreels A2 [11], and VACE 1.3B [24]. As shown, Pika, Vidu, and Kling often focus primarily on the object, resulting in the human face disappearing from the generated frames. Skyreels A2, on the other hand, struggles with producing smooth transitions between frames, leading to lower overall video quality. VACE sometimes fails to capture the intended interaction between the human and the object (left example), and occasionally does not preserve the appearance of the specified object (right example). In contrast, our model consistently maintains strong subject consistency for both the human and the object, while also generating natural and coherent interaction motions between them.

**Human-human & animal-animal customization.** We further provide comparative results for human-human and animal-animal customization tasks in Fig. A3. It can be observed that Pika still suffers from incorrect attention, focusing on hands rather than preserving human identities, and introduces artifacts such as generating three giraffes in the right example. Kling copies lighting from the background of the man image, which contradicts the prompt 'road', and fails to capture the full body of the giraffe, focusing only on the head without depicting the fighting action specified in the prompt, indicating limited prompt adherence. Skyreels A2 fails to represent both subjects and exhibits poor identity preservation. VACE alters the generated human identities and does not follow the fighting prompt in the giraffe example. While Vidu demonstrates relatively better performance, its identity preservation remains suboptimal. In comparison, our model achieves the best identity consistency and prompt adherence, demonstrating superior capability in customized video generation.

# F  More visualization results.

In this section, we present additional visualization results of our model, covering a wide range of subject customization scenarios, including human-object interaction, human-scene interaction,

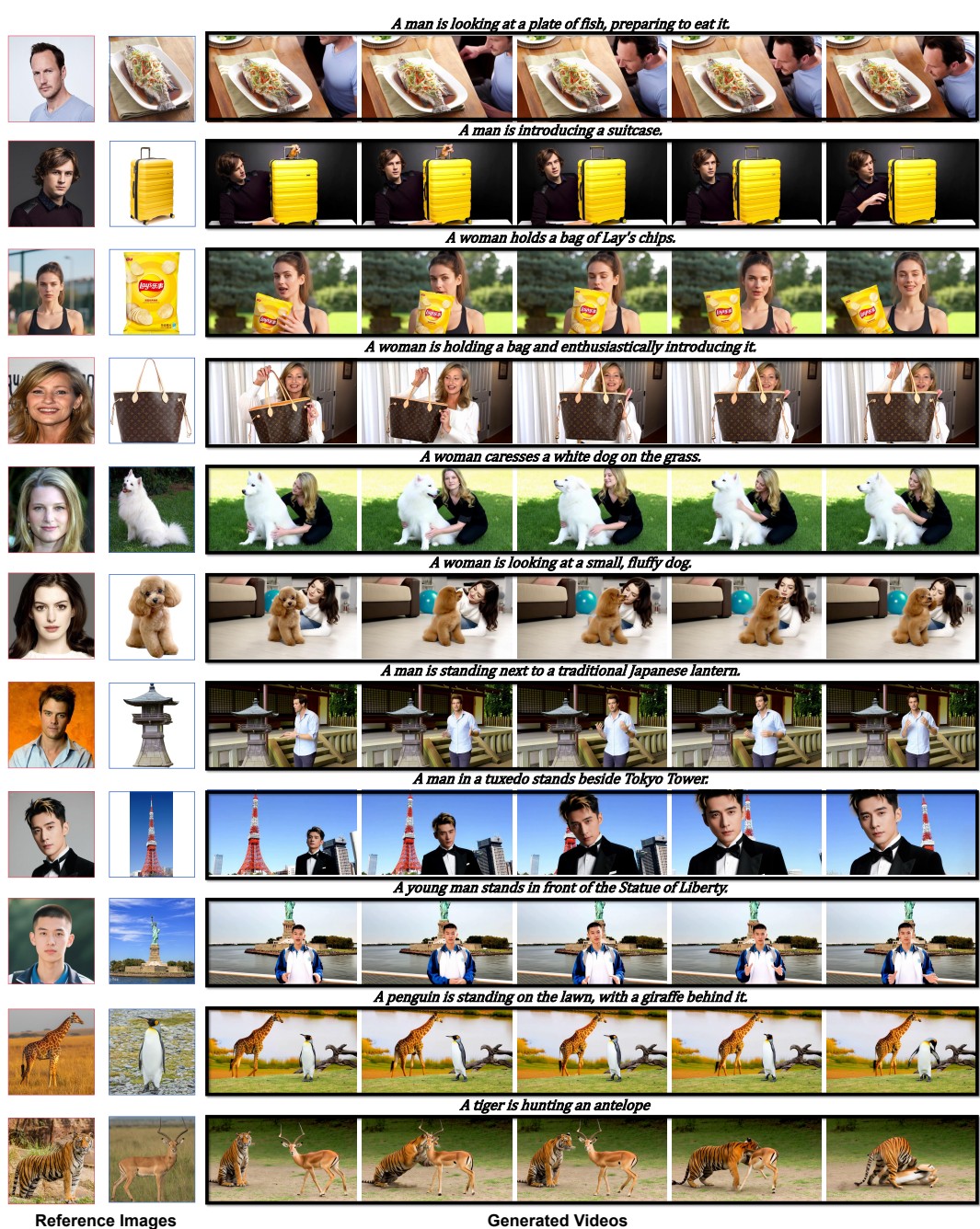

**Reference Images**        **Generated Videos**

Figure A4: More results on multi-subject customization.

human-animal interaction, and animal-animal interaction. We also provide more examples of three-subject customization.

The two-subject customization results are shown in Fig. A4. It can be observed that our model is capable of generating natural and realistic interactions between various types of inputs, demonstrating its potential effectiveness in applications such as advertising and movie production. Furthermore, beyond object interactions, our model can also generate specified subjects within assigned scenes, which is particularly useful for personalized content creation and other creative industries.

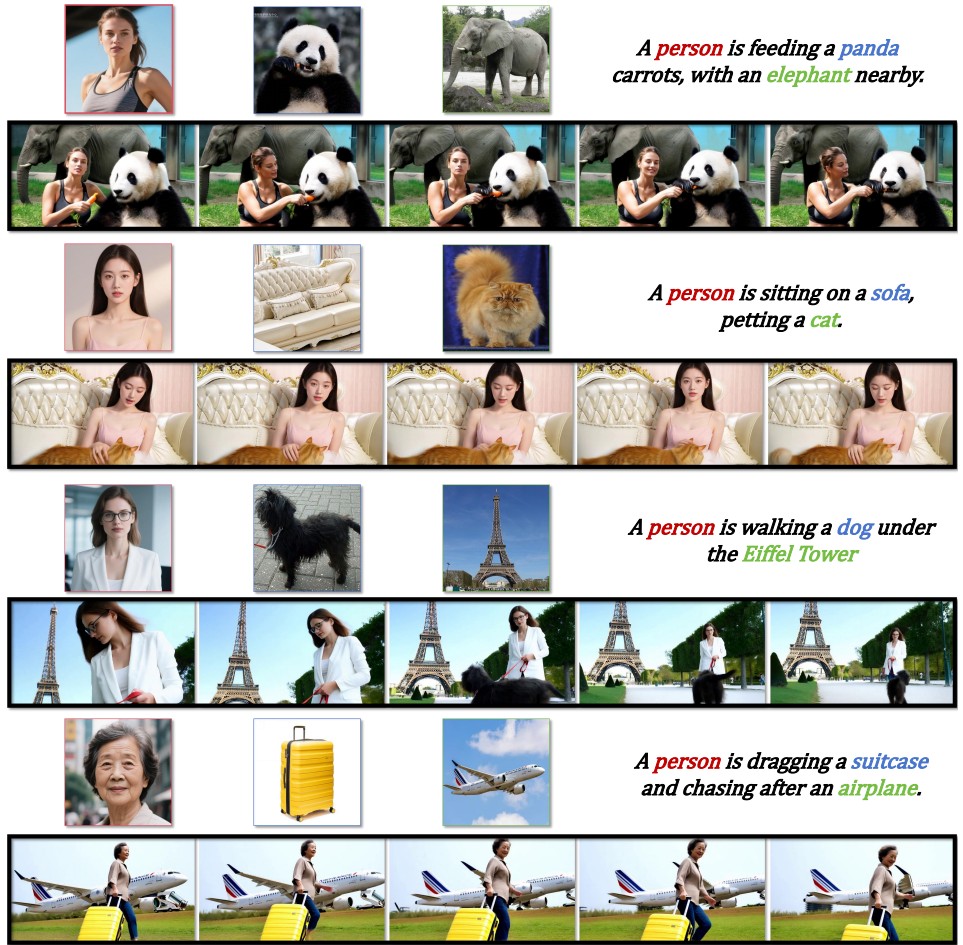

Figure A5: More results on three-subject customization.

Next, we showcase more results of three-subject customization in Fig. A5, featuring diverse combinations such as human-animal-animal, human-object-animal, human-animal-scene, and human-object-object. These results illustrate that our model can effectively handle different combinations of inputs and generate complex interactions among multiple subjects, all while maintaining strong identity preservation. This demonstrates the superior capability of our model in customized video generation for multi-subject scenarios.

## G   Limitations and societal impacts

**Limitations.** Despite the significant advancements introduced by PolyVivid in multi-subject video customization, several limitations remain. First, the quality and controllability of the generated videos are still constrained by the capabilities of the underlying video generation backbone and the pre-trained MLLM and VAE models. Second, while our framework demonstrates strong performance on a variety of subject types and interactions, it may encounter difficulties when handling highly complex scenes involving numerous subjects, intricate backgrounds, or fine-grained interactions that require detailed physical reasoning. Finally, although our MLLM-based data curation pipeline improves subject discriminability, it may still be susceptible to errors in grounding or segmentation, especially in cases of occlusion or ambiguous visual cues, potentially affecting the accuracy of subject alignment and interaction modeling.

**Societal Impacts.** PolyVivid enables more flexible and controllable video generation, which can benefit a wide range of applications, including creative content production, personalized education, digital marketing, and virtual reality experiences. By allowing users to customize videos with specific subjects and interactions, our framework empowers artists, educators, and businesses to efficiently create tailored visual content.

