# OpenReview forum: "PolyVivid: Vivid Multi-Subject Video Generation with Cross-Modal Interaction and Enhancement"
_NeurIPS.cc/2025/Conference — NeurIPS 2025 poster_

### Official Review · Reviewer_mzgG · 2025-06-29

**Clarity:** 3
**Significance:** 4
**Originality:** 3
**Rating:** 5
**Confidence:** 4

**Summary:**

Text-to-video generation has attracted great attention recently. Despite the significant developments, existing text-to-video generation methods still lack fine-grained controllability, especially for multi-subject customization with consistent identity and interaction. To solve the problem, the paper proposes PolyVivid, a multi-subject video customization framework enabling controllable multi-subject video generation. Specifically, PolyVivid utilizes a VLLM-based text-image fusion module to encode subject images and uses a 3D-RoPE-based identity-interaction enhancement module to further enhance identity preservation and enable richer subject interaction. Besides, to efficiently inject text embeddings and image embeddings, PolyVivid proposes an attention-inherited identity injection module. Experiments have shown the effectiveness of the proposed methods.

**Questions:**

Please see weaknesses. Besides, here are some other questions and suggestions below:

* A comprehensive comparative analysis with the backbone model (HunyuanVideo) is required, encompassing both qualitative and quantitative evaluations to assess the relative improvements of the proposed method. On the one hand, whether the approach demonstrates improvements in established evaluation metrics, and on the other hand, whether it enables the generation of previously failed cases that the backbone model could not produce.
* In Figure 6 of the Appendix, the case ``A person is sitting on a sofa, petting a cat." The generated video seems to have a blurry mosaic at the woman's elbow. What could this be due to?
* The paper mentioned that 'existing works lack fine-grained controllability, especially for multi-subject customization with consistent identity and interaction'. I would like to know if the method is able to generate some complex counterfactual relations. For example, "A horse rides on an astronaut."

**Ethical Concerns:**

["NO or VERY MINOR ethics concerns only"]

**Final Justification:**

I have read the authors’ rebuttal, which addressed most of my concerns and provided a more detailed explanation of the shared MM attention mechanism. Although the model still lacks the ability to generate certain counterfactual results due to limitations of the base architecture, the work remains solid at the current research stage. Therefore, I recommend acceptance.

**Limitations:**

yes

**Quality:**

4

**Strengths And Weaknesses:**

Strengths：

* Video generation involving fine-grained interactions between multiple subjects is a critically important and interesting problem in the field.

* The proposed 3D-RoPE mechanism is novel.


Weaknesses:

* The organization of the manuscript has led to an inadequately substantiated experimental component, lacking the depth and rigor required to fully evaluate the proposed approach.
* The sharing MM Attention mechanism illustrated in Figure 3 lacks a detailed explanation in the main text, which significantly confuses the reader, making it difficult to understand the methodological intent when reading Section 4.3

---

> ### Author Rebuttal · Authors · 2025-07-31
>
> Thank you very much for your thorough review and valuable feedback. Your comments have provided us with important guidance for improving our manuscript. Below, we address each of your concerns in detail.
>
> ___
> ### **Q1: About the organization of the manuscript**
>
> Thank you very much for your valuable feedback. We appreciate your suggestion regarding the organization of the manuscript. In the revised version, we will further streamline some of the methodological details and move certain sections to the supplementary material. This will allow us to present a more focused and rigorous experimental section, thereby providing a more comprehensive evaluation of our proposed approach. Thank you again for your constructive comments.
>
> ___
> ### **Q2: Lack of detailed explanation for the sharing MM Attention mechanism**
>
> Thank you very much for your insightful comment. We apologize for any confusion this may have caused.
>
> To clarify, **the sharing MM Attention mechanism in our framework refers to the use of a single set of core parameters for both MM Attention modules and the MM Cross Attention module illustrated in Figure 3**. Specifically, these shared parameters are initialized from the pre-trained weights of Hunyuan-Video. For the Text-Video Interaction, the MM Attention directly utilizes this shared parameter set. In contrast, for the Text-Image MM Attention and the MM Cross Attention, we introduce an additional LoRA (Low-Rank Adaptation) layer for the shared parameters separately. This design enables each module to learn task-specific features through the additional LoRA layers, while still leveraging the knowledge contained in the shared pre-trained weights. By sharing the main set of parameters across modules, the model can converge more quickly during training and requires fewer overall parameters, thus improving both efficiency and performance.
>
> We will update the main text in the revision to provide a more comprehensive and explicit explanation of this mechanism in Section 4.3. Thank you again for your valuable feedback, which will help us improve the clarity and quality of our manuscript.
>
> ___
> ### **Q3: Need for comprehensive comparison with the backbone model**
>
> Thank you very much for your insightful suggestion. We would like to clarify that the original HunyuanVideo model [1] does not support subject-driven customization. However, in our ablation studies, the (*LLaVA + Token-Concatenation*) baseline actually corresponds to our reproduction of the official HunyuanCustom model [2] (official work of HunyuanVideo designed for video customization) for multi-subject customization.
> We have conducted quantitative comparisons between our proposed method and this baseline. As shown in the table below, our approach demonstrates consistent improvements across established evaluation metrics:
>
> | Method         | Facesim ↑ | DINOsim ↑ | CLIP-B ↑ | CLIP-L ↑ | FVD ↓   | Temporal ↑ |
> |----------------|-----------|-----------|----------|----------|---------|------------|
> | Ours           | **0.642**     | **0.623**     | **0.336**    | **0.281**    | **959.74**  | **0.964**      |
> | HunyuanCustom  | 0.628     | 0.615     | 0.328    | 0.271    | 980.56  | 0.960      |
>
> As shown in the table, our method achieves higher scores on Face similarity and DINO similarity, indicating improved identity and appearance consistency. The improvements in CLIP-B and CLIP-L further demonstrate better semantic alignment with the input prompts. Moreover, the better FVD and temporal score value suggest that our generated videos are more temporally coherent and visually realistic. In addition, we have conducted more qualitative comparison with HunyuanCustom [2], where we observe that our method is able to generate videos with more complex subject interactions and maintain fine-grained subject details (e.g., text on the subject), which are often challenging for the baseline model. These results demonstrate the effectiveness and robustness of our approach in both quantitative and practical aspects.
>
> We appreciate your suggestion and will further highlight these comparative results in the revised manuscript to provide a clearer assessment of the relative improvements brought by our method. Thank you again for your constructive feedback.
>
> ___
> ### **Q4: Blurry artifact in generated video (Figure 6, Appendix)**
>
> Thank you very much for your insightful comment. This likely refers to some scarf-like items, as identified by the model, which are draped over the woman's elbow. However, since only the main part of these items is occluded by the cat, they appear to resemble a blurry mosaic.
>
> ___
> ### **Q5: Ability to handle complex counterfactual relations**
>
> Thank you very much for your valuable comment. Our model is built upon HunyuanVideo, and therefore, its ability to model complex counterfactual relations is inherently limited by the base model. We have tested both HunyuanVideo [1] and Wan [3] on this prompt, and neither is able to generate the correct video (both produce "An astronaut rides on a horse"). Our method also finds it challenging to generate such counterfactual cases (although it can generate the correct "An astronaut rides on a horse"). We believe that as the capabilities of base models continue to improve, applying our method to a more powerful base model holds great promise for addressing these cases.
>
> [1] Kong W, Tian Q, Zhang Z, et al. Hunyuanvideo: A systematic framework for large video generative models[J]. arXiv preprint arXiv:2412.03603, 2024.
>
> [2] Hu T, Yu Z, Zhou Z, et al. Hunyuancustom: A multimodal-driven architecture for customized video generation[J]. arXiv preprint arXiv:2505.04512, 2025.
>
> [3] Wan T, Wang A, Ai B, et al. Wan: Open and advanced large-scale video generative models[J]. arXiv preprint arXiv:2503.20314, 2025.

---

> ### Author Response · Authors · 2025-08-06
>
> Dear Reviewer mzgG,
>
> Thank you again for your time in reviewing our paper. We would appreciate it if you could let us know whether our response has addressed your concerns, or if there are any remaining issues we should clarify.
>
> Best regards,
>
> Authors of PolyVivid

---

> ### Comment · Reviewer_mzgG · 2025-08-07
>
> I have read the authors' rebuttal, which addressed most of my concerns. Therefore, I will maintain my positive score.

---

> > ### Author Response · Authors · 2025-08-07
> > **Thank Reviewer mzgG for the Comments**
> >
> > Dear Reviewer mzgG,
> >
> > Thank you very much for your recognition of our work. We sincerely appreciate your valuable comments and constructive suggestions, which are very helpful for improving our manuscript. We are grateful for the time and effort you have devoted to reviewing and providing feedback on our paper.
> >
> > Best regards,
> >
> > The authors of Polyvivid

---

### Official Review · Reviewer_sm4D · 2025-06-30

**Clarity:** 4
**Significance:** 3
**Originality:** 3
**Rating:** 5
**Confidence:** 3

**Summary:**

The paper proposes PolyVivid, a novel inference-time multi-subject video customization framework composed of several components. It introduces a design that leverages VLLMs for semantic understanding of both reference images and texts, along with a novel 3D-RoPE-based identity interaction enhancement module, which effectively preserves the sequential relationship between text and images through bi-directional fusion. Additionally, the paper presents an identity injection attention module featuring a new adaptation method for handling image tokens, and a data construction pipeline that consider subjects appearing only briefly in the video. This pipeline demonstrates superior performance even compared to recent closed-source models, such as Pika and Kling.

**Questions:**

**Q1**. While the qualitative samples suggest that identity preservation is well handled, some samples give a slight copy-and-paste impression, e.g., Figure 5 in the supplementary material. Was there any attempt to filter out overly similar segmentations in the dataset or to refine the data construction process for selecting frames?

**Q2**.. How does the model perform on more outlier-like concepts (e.g., the monster toy in DreamBooth) or with highly similar subjects, such as two male individuals? Additionally, I’m curious whether the model can handle more complex compositions—beyond side-by-side placements—such as a scene featuring a can with the monster toy illustrated on it. (This isn’t meant as a negative comment, I understand that video models still have limitations in editability.)

**Ethical Concerns:**

["NO or VERY MINOR ethics concerns only"]

**Final Justification:**

I have read the authors' responses, and my concerns about this work have been fully addressed. Considering its practical applicability, I intend to recommend acceptance.

**Limitations:**

yes

**Paper Formatting Concerns:**

There is no issue with the formatting.

**Quality:**

3

**Strengths And Weaknesses:**

**S1**. The paper is easy to read and follow, and the presentation is clear. The supplementary material is also well organized.

**S2**. Clique-based subject consolidation is interesting. This simple design effectively improves the robustness of multi-subject personalization datasets.

**S3**. 3D-RoPE-based identity-interaction enhancement module is a novel and straightforward design for text-image fusion within the attention module. This structure presents a promising direction for future research in personalization, or in broader in-context image or video generation.

**S4**. The method also demonstrated superior performance compared to closed-source models such as Kling and Pika.

**W1**. (minor) I think the model name Keling should be Kling. (Kling appears to be the official name)

**W2**. Image and text fusion via VLLMs is not a novel idea. I think citing [1] and [2] would be helpful.

[1] Song et al., MoMA: Multimodal LLM Adapter for Fast Personalized Image Generation, ECCV 2024

[2] Li et al., BLIP-Diffusion: Pre-trained Subject Representation for Controllable Text-to-Image Generation and Editing, NeurIPS 2023

---

> ### Author Rebuttal · Authors · 2025-07-31
>
> We sincerely thank you for your valuable comments and suggestions, which have greatly helped us improve our work. Below, we provide detailed responses to each of your points.
> ___
> ### **Q1: Correction of Model Name ("Keling" → "Kling")**
>
> Thank you very much for your valuable suggestion. We will make the corresponding revisions in the revised manuscript.
>
> ___
> ### **Q2: Novelty of Image and Text Fusion via VLLMs; Citation of Related Works**
>
> Thank you very much for your valuable suggestion. Our main contribution lies in designing a structured template for video customization models, which helps the video model better understand the relationship between the subject image and the text. We will cite these works in the revised manuscript to make the related work section more rigorous. Thank you again for your helpful comments.
> ___
> ### **Q3: Copy-and-Paste Impression in Qualitative Samples; Data Processing for Identity Preservation**
>
> Thank you very much for your valuable comment. In our data construction process, we used images of the same subject from other video clips as conditions, rather than always using the main subject image from the same video. This approach helps to alleviate the hard-copy issue. However, when the given subject image matches the environment specified by the prompt, the model may still copy most of the features from the image into the generated video, as this not only preserves the subject's characteristics but also closely follows the text prompt.
>
> ___
> ### **Q4: Handling Outlier Concepts, Highly Similar Subjects, and Complex Compositions**
>
> Thank you very much for your thoughtful comments and for raising these important questions.
>
> Benefiting from the strong generative capabilities of our base model, HunyuanVideo, as well as the robust generalization ability of our proposed method, we have experimentally verified that our approach can successfully handle outlier-like concepts such as the "monster toy" in DreamBooth, as well as highly similar subjects, such as two male individuals. Furthermore, our method is also capable of generating more complex compositions beyond simple side-by-side arrangements. For example, we are able to generate scenes like "a can with the monster toy illustrated on it." This compositional ability can be partially observed in the clothing transfer case shown in Fig. 4, where the model demonstrates the capacity to combine subject attributes in a more integrated manner.
>
> Of course, we acknowledge that for some highly counter-intuitive or extremely challenging cases, there may still be limitations due to the current capabilities of video generation models. We greatly appreciate your suggestions, and in the revised version of our paper, we will include additional representative cases to better illustrate the strengths and current limitations of our approach. Thank you again for your valuable feedback.

---

> > ### Comment · Reviewer_sm4D · 2025-08-03
> >
> > Thanks for the authors' rebuttal that addresses many of my concerns. I will keep my positive rating.

---

> > > ### Author Response · Authors · 2025-08-04
> > > **Thank Reviewer sm4D for the Comments**
> > >
> > > Dear Reviewer sm4D,
> > >
> > > Thank you very much for your recognition of our work. We sincerely appreciate your insightful suggestions and will carefully incorporate them into our revised manuscript. We are grateful for the time and effort you have dedicated to reviewing and discussing our paper.
> > >
> > > Best regards,
> > >
> > > The authors of Polyvivid

---

### Official Review · Reviewer_Mbak · 2025-07-01

**Clarity:** 3
**Significance:** 4
**Originality:** 4
**Rating:** 5
**Confidence:** 3

**Summary:**

This paper presents a method for multi-subject personalized video generation, built on top of HunyuanVideo (a video diffusion model). To enhance identity preservation and interaction control, the authors introduce three key components:

- A new multi-subject customization dataset, preprocessed using MLLM, SAM, CLIP, and a clique-based subject grouping strategy to support diverse and coherent interactions.

- A 3D-RoPE-based identity interaction module, which encourages information exchange between embeddings from the VAE (capturing detailed identity) and a VLLM (LLaVA) responsible for interaction modeling.

- An attention-inherited identity injection mechanism for more effective subject-specific conditioning throughout the video sequence.

**Questions:**

1. Can the model reliably disambiguate identities when subjects belong to the same category?

For example, in a prompt like ``a man on the left, and a man on the right``, does the model consistently assign the correct identity across frames? I reviewed Figure 3 of the appendix (the two-giraffe example), but it does not provide sufficient evidence to assess whether the model can reliably control identity in multi-subject scenarios where all subjects belong to the same category.

2. What are the limitations of the model with respect to subject scale(size) and visibility?

 Can it simultaneously handle very small objects (e.g., a ring) alongside full-body subjects? Additionally, how does the model behave when a subject should be occluded or temporarily disappears from prompt? It's because I suspect the model tend to maintain the subjects visible in frames across video.

3. Token Dimensions

In Equations (5) and (6), both the <image> token embedding $z_{T,I}$ and the identity token $z_I$ are said to have spatial dimensions of $w \times h$. However, if my understanding is correct, $z_{T,I}$ comes from LLaVA and $z_I$ from the Hunyuan VAE, which may produce different token grid sizes. Are these embeddings resized or projected to enforce matching spatial shapes?

**Ethical Concerns:**

["NO or VERY MINOR ethics concerns only"]

**Final Justification:**

I've read authors rebuttal, and it effectively resolved my concerns on ambiguity in the method part and clarified the scope that model can handle (limitation) clearly. Given its practical applicability, I'm inclined to recommend accept.

**Limitations:**

The model may struggle to handle more than three reference images effectively.

**Quality:**

3

**Strengths And Weaknesses:**

Strengths:
- The paper is well written and easy to follow.

- The generated videos show better performance in terms of visual quality, identity preservation, and prompt alignment, compared to baselines (including commercial models) —quantitatively and qualitatively.

- The use of 3D RoPE embeddings for identity interaction enhancement is novel and effective, contributing significantly to performance.


Weaknesses:
- Since the model accepts a clique length longer than one-third of the total sequence, it may struggle to handle more than three subjects.

- The model may fail to disambiguate identities when subjects are described using similar word, potentially leading to identity permutation.

For example, ``a man is standing next to a chair, and an another man is sitting on the chair.`` It will be difficult to assign each identity, as it use same natural language as description word.

- Although the qualitative results are strong, the paper lacks a detailed evaluation of video quality. Metrics such as those from VBench (e.g., motion consistency) or human preference studies are not reported, so, it's difficult to determine whether fine-tuning introduces any trade-offs in visual quality.

---

> ### Author Rebuttal · Authors · 2025-07-31
>
> We sincerely thank you for your valuable comments and suggestions, which have greatly helped us improve our work. Below, we provide detailed responses to each of your points.
>
> ___
> ### **Q1: About how to handle more than three subjects**
>
> Thank you very much for your careful reading and insightful comment. We apologize for any confusion caused by our description. We would like to clarify that in our clique-based subject consolidation algorithm, **the "total sequence" refers to the total number of frames in the video (i.e., video length), rather than the total number of subject images**. Specifically, the threshold for the maximum clique is set to one-third of the total number of frames, not the number of detected subject images. This design ensures that each identified subject appears in a sufficiently large portion of the video, thereby filtering out subjects that only appear briefly and improving the robustness of multi-subject detection and segmentation. Therefore, our method can handle more than four subjects in a video.
>
> We hope this clarifies the intention and effect of our algorithm. Thank you again for your valuable feedback. Additionally, we will include additional cases with four subjects in our revision to further demonstrate the scalability and effectiveness of our method in handling more than three subjects.
>
> ___
> ### **Q2: About subjects described using similar word**
>
> The presence of similar subjects indeed poses a significant challenge for existing multi-subject video customization methods. To address this issue, we have designed three modules, spanning from data to model level, to enhance the discrimination of similar subjects.
>
> **(1) At the data level,** we employ a multi-modal large language model (MLLM) to distinguish similar objects mentioned in the text. Furthermore, we utilize a clustering algorithm to prevent confusion among different objects, which is not achievable in the data pipelines of previous approaches.
>
> **(2) At the model level,** we first leverage LLaVA to facilitate the model’s ability to differentiate similar subjects. Specifically, we introduce identity prompts to distinguish between similar subjects; for example, LLaVA receives prompts such as “a man looks like \<image1\>, the other man looks like \<image2\>,” enabling the model to explicitly differentiate between different subjects.
>
> **(3) In addition,** our 3D interactive ROPE mechanism places the subject images near their corresponding textual tokens, thereby strengthening the identity of each subject and further improving subject discrimination.
>
> Following your suggestions, we have conducted experiments to validate our model’s ability to handle similar subjects, where we achieve a Face-Sim of 0.632 and DINO-sim of 0.618. We will include relevant illustrative examples in the revision.
>
> ___
> ### **Q3: More metrics in Vbench**
>
> Thank you very much for your thoughtful suggestion. In our paper, the **temporal consistency** metric we reported corresponds to the "subject consistency" and "background consistency" metrics in VBench. Following your advice, we have further evaluated our method using additional VBench metrics, specifically **motion consistency** and **image quality**.  In addition, we conducted a **human preference study** with 32 volunteers, who ranked the generated videos based on subject consistency, text alignment, and overall video quality. The average rankings (lower is better), along with the Vbench metrics, are reported in the following table.
>
> | Method   | Temporal consistency ↑ | Motion consistency ↑ | Image quality ↑ | Human preference ↓ |
> |----------|-----------------------|----------------------|-----------------|--------------------|
> | Pika     | 0.942                 | 0.995                | 0.729           | 3.00               |
> | SkyReels | 0.943                 | 0.994                | 0.731           | 5.41               |
> | VACE     | *0.966*               | 0.994                | *0.732*         | 4.28               |
> | Kling    | 0.934                 | 0.994                | 0.724           | 3.12               |
> | Vidu     | **0.970**             | **0.997**            | 0.695           | 3.47               |
> | Ours     | 0.964                 | *0.995*              | **0.751**       | **1.72**           |
>
> As shown in the table, our method achieves highly competitive temporal and motion consistency, on par with the best-performing baselines. More importantly, our approach achieves the highest image quality score among all methods, suggesting that our model produces the best perceptual quality of the generated videos. Furthermore, our method receives the most favorable human preference ranking, indicating that users consistently prefer our results over other baselines in terms of subject consistency, text alignment, and overall video quality. We will include these additional quantitative results and analyses in the revised version of our paper to provide a more comprehensive evaluation. Thank you again for your valuable feedback.
>
> ___
> ### **Q4: Experiment on subject scale (size) and visibility**
>
> **Our method is capable of handling small objects as well as occluded or intermittently appearing objects**. This is because, in our data processing pipeline, we do not strictly constrain the size of the extracted subject images, and we retain objects that appear for more than one third of the total video duration. Therefore, our model is theoretically able to process small and occluded/disappearing objects. In practice, the capability in generating occluded/disappearing objects is demonstrated in Supplementary Figure 5 (example 8), where a person blocks the Tokyo Tower, and in Supplementary Figure 6 (example 3), where the dog only appears in the latter half of the video. Additionally, we have validated that our method can handle cases where an object first appears and then disappears, as well as extremely small objects such as rings. We will include more such examples in the revision. Thank you for your suggestion.
>
> ___
> ### **Q5: About Token Dimensions**
>
> Thank you very much for your insightful comments. Your understanding is indeed correct. We have resized the images input to the VAE to ensure alignment with the grid size used by LLaVA. We will clarify this point in the revised manuscript. We sincerely appreciate your valuable suggestion.

---

> > ### Comment · Reviewer_Mbak · 2025-08-05
> >
> > Thank you to the authors for the rebuttal, which addressed many of my concerns. I will maintain my positive assessment.

---

> > > ### Author Response · Authors · 2025-08-05
> > > **Thank Reviewer Mbak for the Comments**
> > >
> > > Dear Reviewer Mbak,
> > >
> > > Thank you very much for your recognition of our work. We sincerely appreciate your insightful suggestions and will carefully incorporate them into our revised manuscript. We are grateful for the time and effort you have dedicated to reviewing and discussing our paper.
> > >
> > > Best regards,
> > >
> > > The authors of Polyvivid

---

### Official Review · Reviewer_5tse · 2025-07-04

**Clarity:** 3
**Significance:** 2
**Originality:** 2
**Rating:** 2
**Confidence:** 5

**Summary:**

The paper presents a method for multi-reference condition to a text-to-video model. The method encodes both the video textual caption and the reference images into llava embedding space. The authors propose to modify the rope module in order to improve the interaction between the llava embeddings and the VAE embeddings of the image references. The authors propose an additional modification to the attention modules aiming to improve the contextualization of the image embedding with the video embedding sequence.

**Questions:**

* The Attention-inherited Identity Injection is not entirely clear. While Figure 3 shows the proposed modification, it can be helpful to also specify more clearly in the text what is the difference compared to the baseline architecture. Specifically, add to Equations 7-8 the entire flow of the attention as shown in the Figure.

**Ethical Concerns:**

["NO or VERY MINOR ethics concerns only"]

**Limitations:**

The authors don't specifically addressed potential negative societal impact of their work. However, I think the limitations of their method are not fundamentally different from limitations of similar papers accepted to neurips and similar conferences.

**Quality:**

2

**Strengths And Weaknesses:**

Strengths:
* Relevant works are discusses.
* The results seem appealing.

Weaknesses:
* Overall, the pipeline seems overcomplicated. I think that most of the new design choices that the paper introduces are unnecessary for properly integrating multi reference condition into the video model.
* The VLLM-based Text-image Fusion (Sec 4.1) seem redundant to me, and also very specific to Hunyuan which uses llava as opposed to T5, this also reduced efficiency. To my understanding, all the experiments in the ablation study feed in the images into llava, so this component is not motivated + may contaminate the other experiments in the ablation study. I am missing the simple baseline of text-only llava embeddings (the caption may refer to the reference images!) and concatenation of the image VAE embedding to the sequence, without any other modifications to the overall architecture.
* It's unclear to me why a special rope is needed, what is the problem with simply treating the reference images as additional frames and use the existing rope module of the video model?
* The authors claim that simply concatenating the image vae embedding to the vae embedding sequence has a limitation as distant frames will be affected less by the condition image. However, I am not convinced that this speculation is valid. It is also worth mentioning that the authors use Hunyuan which uses DiT with full sequence attention, so the reference image directly interacts with each frame in the video. Indeed, the ablation study shows that the identity preservation increases with simple token concatenation.

---

> ### Author Rebuttal · Authors · 2025-07-31
>
> We sincerely thank the reviewers for their valuable comments and suggestions, which have greatly helped us improve our work. Below, we provide detailed responses to each point.
>
> ___
>
> ### **Q1: The necessity of each proposed module**
>
> Thank you very much for your valuable feedback and for sharing your perspective on the pipeline design. Our intention in introducing these design choices was to ensure a more comprehensive integration of multi-subject conditions into the video model and to address specific challenges observed in our preliminary experiments. We would also like to note that, despite these additional modules, our overall architecture remains straightforward and does not introduce significant computational overhead, as evidenced by the nearly identical inference time (4x H20 3.6 min) compared to the simple concatenation baseline (4x H20 3.5 min). We explain the motivation and necessity of each module in detail below.
>
> -  **The VLLM-based Text-image Fusion** is introduced to help the model understand the correlation between the text prompt and subject images. The model without this fails to distinguish which subject image corresponds to which text word. This is extremely serious when the subject images are similar items (e.g., two human, two cats). We have conducted additional experiments to validate this in the response for Q2.
>
> - **Text-image interaction 3D-ROPE:** The standard ROPE suffers from long-term decay, resulting in insufficient identity information for frames farther from the reference image. Our text-image interaction 3D-ROPE, in combination with the identity injection and text-video interaction modules (as shown in Fig. 3(c)), addresses this issue by enhancing the propagation of identity information and improving prompt following throughout the video sequence (More details are in the response for Q3).
>
> - **The Identity Injection and Text-Video Interaction Modules** are essential for enhancing the model’s ability to capture identity information and to follow the text prompt accurately. Compared to simple token concatenation, our approach effectively mitigates the long-term decay problem, ensuring that identity and prompt information are preserved throughout the video sequence. This leads to significantly improved multi-subject generation capability and better consistency between the generated content and the input text and image conditions.
>
> ---
>
> ### **Q2: About the VLLM-based Text-image Fusion**
>
> Thanks for your suggestion. The text-image Fusion Module is necessary for multi-subject customization. Since the model needs to distinguish between different input subjects, i.e., which image correspond to which words. If there is not the Text-image Fusion module, the model is impossible to distinguish similar objects. (e.g., there the two subjects are 'a dog'). Therefore, we do not include the mentioned model with text-only llava and image concatenation in the ablation study due to its inherent limitation. Following your suggestion, we additionally conducted an ablation study of this model, the results are shown below.
>
> | Model           | Face-sim ↑ | DINO-sim ↑ | CLIP-B ↑ | CLIP-L ↑ | FVD ↓    | Temporal ↑ |
> |----------------|:----------:|:----------:|:--------:|:--------:|:--------:|:----------:|
> | **Ours** | **0.642**  | **0.623**  | **0.336**| **0.281**| **959.74**| **0.964**  |
> | Ablated model      | 0.543      | 0.532      | 0.330    | 0.267    | 1132.12  | 0.959      |
>
> It can be seen that the model without text-image Fusion results in a poor identity consistency due to the limitation in distinguishing different identities, validating the necessity of it.
>
> ___
>
> ### **Q3: About the text-image interaction 3D-RoPE**
>
> Thank you very much for your insightful question. The standard ROPE positional encoding is known to suffer from long-term decay [1], which means that frames farther from the reference image receive less conditional information. Simply treating the reference image as an additional frame and using the existing ROPE can therefore result in insufficient identity information for distant frames, leading to identity degradation.
>
> Our text-image interaction 3D-ROPE addresses this issue by enhancing both the identity information from the text prompt and the interaction between the reference image and video frames. This enables our model to achieve better identity preservation and prompt following, with the combination of the identity injection and text-video interaction module in Fig.3 (c).
>
> ___
> ### **Q4: About the long-term decay of the reference image in simple concatenation and the ablation study**
>
> Thank you very much for the comments and for raising these important points.
>
> **Regarding distant frames,** We appreciate your insights regarding the use of simple token concatenation in DiT-based models such as Hunyuan, which employs ROPE as the positional embedding. While it is true that full sequence attention allows the reference image to interact with each frame, prior studies have shown that ROPE exhibits long-term decay [1], meaning that frames further away from the condition image tend to have weaker correlations with it. This can lead to identity degradation in the latter frames, as they do not receive sufficiently accurate identity information through simple token concatenation alone.
>
> To further validate this problem, we conducted experiments using direct token concatenation and standard 3D-ROPE. Specifically, we measured the similarity between each segment of frames (from $t\times10$ to $(t+1)\times10$) and the reference image as time t increases, and compared these results with those obtained using our proposed method in the following table:
>
> | Time t         | 0      | 1      | 2      | 3      | 4      | 5      | 6      | 7      | 8      | 9      | mean   | std    |
> |----------------|:------:|:------:|:------:|:------:|:------:|:------:|:------:|:------:|:------:|:------:|:------:|:------:|
> | Simple Concat  | 0.649  | **0.644** | **0.645** | **0.642** | 0.636  | 0.621  | 0.626  | 0.607  | 0.624  | 0.609  | 0.630  | 0.015 |
> | Ours           | **0.651** | **0.644** | 0.638  | 0.640  | **0.641** | **0.645** | **0.643** | **0.647** | **0.639** | **0.646** | **0.643** | **0.004** |
>
>
> **Regarding the ablation study,** we agree that simple concatenation achieves relatively good identity preservation. However, our results demonstrate that it still falls short of the performance achieved by our proposed identity injection mechanism. Furthermore, in the context of video customization, prompt following ability is also crucial. Our experiments show that the simple token concatenation approach yields a lower CLIP score compared to our method, indicating inferior prompt alignment.
>
> In summary, while simple token concatenation provides a reasonable baseline, our identity injection mechanism offers more robust identity preservation and better prompt following, addressing the limitations of existing approaches. We are grateful for your feedback, which has allowed us to clarify and further emphasize the advantages of our proposed design.
> ___
>
> ### **Q5: About the Attention-inherited Identity Injection**
> Thank you for your helpful suggestion. We will revise the manuscript to clarify the differences between our Attention-inherited Identity Injection and the baseline architecture. Specifically, we will update Equations 7–8 to explicitly show the entire flow of the attention as follows:
>
> $$
> \hat{z}_I, \hat{z}_T = \hat{FFN}(MM\text{-}Attn_1(z_I, z_T))
> $$
>
> $$
> z' = Cross\text{-}Attn(\hat{W}_q^V(z), \hat{W}_k^V(\hat{z}_I), \hat{W}_v^V(\hat{z}_I))
> $$
>
> $$
> \hat{z} = z + FC(\hat{FFN}(z'))
> $$
>
> $$
> z_{out}, z_{T, out} =MM\text{-}Attn_2(\hat{z}, \hat{z}_T)
> $$
>
> where $MM\text{-}Attn_1$ is the MM Attention for text-image interaction and $MM\text{-}Attn_2$ is the MM Attention for text-video interaction. We hope this will make our method clearer. Thank you again for your valuable feedback.
>
> ___
> ### **Q6: About the potential negative societal impact**
> Thank you for your comment and for raising this important point. We acknowledge that we did not specifically discuss the potential negative societal impacts of our work in the current version of the paper. We agree that, similar to other works in this area, our method may share common limitations, such as potential misuse for generating misleading or inappropriate content. We will add a discussion on these societal impacts and possible mitigation strategies in the revised version, in line with the standards of NeurIPS and similar conferences. Thank you again for your valuable feedback.
>
> [1] Su J, Ahmed M, Lu Y, et al. Roformer: Enhanced transformer with rotary position embedding[J]. Neurocomputing, 2024.

---

> ### Author Response · Authors · 2025-08-06
>
> Dear Reviewer 5tse,
>
> Thank you again for your time in reviewing our paper. We would appreciate it if you could let us know whether our response has addressed your concerns, or if there are any remaining issues we should clarify.
>
> Best regards,
>
> Authors of PolyVivid

---

> ### Author Response · Authors · 2025-08-08
>
> Dear Reviewer 5tse,
>
> We sincerely appreciate the time and effort you have devoted to reviewing our paper. As the discussion phase will end in one day, we kindly ask if you could let us know whether our response has addressed your concerns, or if there are any remaining issues we should clarify. Your feedback is very important to us.
>
> Thank you again for your consideration.
>
> Best regards,
>
> Authors of PolyVivid

---

### Decision · Program_Chairs · 2025-09-17

**Decision:**

Accept (poster)

**Comment:**

The paper proposes a multi-subject video customization framework for generating identity-consistent videos with multiple subjects based on HunyuanVideo. Reviewers acknowledge several strengths including the novel 3D-RoPE mechanism for identity-interaction enhancement, strong experimental results outperforming commercial baselines, clear presentation, and a novel clique-based data construction approach for filtering inconsistent detections and ensuring subject consistency across video frames.

The weaknesses noted by reviewers include insufficient clarity on technical details, insufficient evaluation, questions about the model's ability to handle complex or ambiguous scenarios, and unclear necessity of certain components (particularly VLLM fusion). Authors provided additional experiments, clarifications, and additional VBench/human evaluation metrics. Three reviewers explicitly acknowledged that their concerns were addressed and recommended clear accept, while Reviewer 5tse did not engage in discussion.

The Area Chair agrees with the engaged reviewers and recommends acceptance. Authors are advised to revise based on reviewers' comments to make the final version more solid. Additionally, kindly discuss the generalizability of the VLLM-based fusion approach beyond the current implementation, addressing concerns that it appears very specific to HunyuanVideo's architecture.